# Isolation of Haustorium Protoplasts Optimized by Orthogonal Design for Transient Gene Expression in *Phelipanche aegyptiaca*

**DOI:** 10.3390/plants13152163

**Published:** 2024-08-05

**Authors:** Xiaojian Zeng, Xiaolei Cao, Qiuyue Zhao, Siyuan Hou, Xin Hu, Zheyu Yang, Tingli Hao, Sifeng Zhao, Zhaoqun Yao

**Affiliations:** Key Laboratory at the Universities of Xinjiang Uygur Autonomous Region for Oasis Agricultural Pest Management and Plant Protection Resource Utilization, Agriculture College, Shihezi University, Shihezi 832003, China; cengxiaojian@stu.shzu.edu.cn (X.Z.); tulanduocxl@sina.com (X.C.); zhaoqiuyue1@stu.shzu.edu.cn (Q.Z.); housiyuanqaz@sina.com (S.H.); huxin@stu.shzu.edu.cn (X.H.); yangzheyu@stu.shzu.edu.cn (Z.Y.); tingli.hao@foxmail.com (T.H.)

**Keywords:** *Phelipanche aegyptiaca*, haustorium, protoplasts’ isolation, transient transformation

## Abstract

The efficient protoplast transient transformation system in plants is an important tool to study gene expression, metabolic pathways, and various mutagenic parameters, but it has not been established in *Phelipanche aegyptiaca*. As a root parasitic weed that endangers the growth of 29 species of plants in 12 families around the world, there is still no good control method for *P. aegyptiaca*. Even the parasitic mechanisms of *P. aegyptiaca* and the related genes regulating parasitism are not yet understood. In this study, by comparing the factors related to protoplast isolation and transfection, we developed the optimal protocol for protoplast isolation and transfection in *Phelipanche aegyptiaca* haustorium. The optimal protoplast yield and activity were 6.2 × 10^6^ protoplasts/g fresh weight [FW] and 87.85%, respectively, by using 0.5 mol/L mannitol, enzyme concentrations of 2.5% cellulase R-10 and 0.8% Macerozyme R-10 at 24 °C for 4 h. At the same time, transfection efficiency of protoplasts was up to 78.49% when using 30 μg plasmid, 40% polyethylene glycol (PEG) concentration, 24 °C incubation temperature, and 20 min transfection time. This is the first efficient protoplasts’ isolation and transient transformation system of *Phelipanche aegyptiaca* haustorium, laying a foundation for future studies on the gene function and mechanisms of haustorium formation in parasitic plants.

## 1. Introduction

The *Phelipanche aegyptiaca* is an annual chlorophyll-free obligatory root-parasitic weed in the family Orobanchaceae, belonging to the *Phelipanche*. This parasitic weed (*P. aegyptiaca*) seriously endangers global agricultural production and is mainly distributed in the Middle East, Africa, the Mediterranean, China, and other semi-arid regions of the world. It parasitizes nearly 29 species from 12 different plant families, such as tomato, potato, melon, bean, pepper, tobacco, and carrot, affecting crop yield and quality, and causing significant losses to global agricultural production [1,2,3]. In 2018, it was estimated that global annual losses owing to *Orobanche* damage were USD 1.3–2.6 billion [4]. Currently, there are no effective preventive measures against this weed. Haustorium, a critical organ in *P. aegyptiaca*, serves as a bridge between host and parasite, facilitating the acquisition of water, nutrients, RNA, proteins, and hormones from the host [5]. Therefore, preventing haustorium formation or its connection with the host will be a fundamental method to control *P. aegyptiaca*.

Recent research indicates that haustorium-repressing factors, particularly nitrogen, can disrupt parasite–host interactions by promoting abscisic acid (ABA) levels, resulting in the repression of genes essential for haustoria development in *Phtheirospermum japonicum* [6,7]. Additionally, quinone-type HIFs likely affect CARD1 activation, a membrane-bound kinase, which triggers YUCCA3-mediated auxin biosynthesis at the haustorium apex, potentially leading to haustorium formation [5]. However, the limited understanding of haustorium-specific signaling pathways and gene interactions in *P. aegyptiaca* has hindered studies on haustorium formation.

Although there is information about stable genetic transformation and plant regeneration of *P. aegyptiaca*, it is mostly related to host, which may interfere with the study of *P. aegyptiaca* gene function. Transient protoplasts’ expression is a validated approach for researching gene function and the signaling pathway, effectively providing insights into molecular mechanisms in plant immunity, hormone signal transduction, and physiological processes, including growth and development. Researchers have used this approach to screen synthetic promoters in poplar leaf mesophyll protoplasts, as demonstrated by Yang et al. [8], and to uncover the transcriptional regulation of *BrpHMA2* under cadmium stress in *Brassica parachinensis* protoplasts [9]. Similarly, high-efficiency, regeneration-free transient expression in haustorium protoplasts of *P. aegyptiaca* could support an efficient system for studying genes and signaling pathways specific to haustoria. The key to the successful transient expression in protoplasts lies in the efficient digestion system for high-yield, high-activity haustorium protoplasts, and an effective transient transformation system of protoplasts. The present methods for plant tissue protoplasts’ digestion mainly include mechanical, chemical, and enzymatic methods, with the enzymatic method being notably efficient [10]. However, there have been no effective and sustainable protocols for *P. aegyptiaca* protoplasts’ digestion [11].

The PEG-mediated method is widely adopted for plant protoplasts’ transient transformation for its efficiency, cost effectiveness, and lack of specialized equipment requirements [12]. Zhang’s work with cotton root protoplasts, achieving an 80% transformation efficiency, exemplifies its potential [13]. However, established PEG-mediated systems predominantly focus on model plants and major crops, with transformation parameters varying across different species and even organs within the same species.

This study will focus on the isolation of haustorium protoplasts from *P. aegyptiaca* by modulating key factors, including enzyme solution composition, digestion conditions, and centrifugation speed. Moreover, by evaluating temperature, incubation time, PEG concentration, and plasmid content on transformation efficiency, we propose an advanced and efficient transformation protocol for haustorium protoplasts from *P. aegyptiaca*. This research will offer a robust method for isolating *P. aegyptiaca* protoplasts, facilitating molecular research on the haustorium of *P. aegyptiaca*.

## 2. Materials and Methods

### 2.1. Plant Materials 

The *P. aegyptiaca* seeds used in this experiment were all collected from tomato fields in Jimusar County, Xinjiang province, China in 2020 [14]. 

### 2.2. Materials for Protoplasts’ Isolation of P. aegyptiaca Haustorium

The seed were surface-sterilized with 2% *v*/*v* sodium hypochlorite in water for 15 min and 75% *v/v* ethanol in water for 3 min, and then the seed were washed five times with sterile water to remove the residual sterilant. The sterilized seeds were placed in 24-well plate and preconditioning with sterile water in a darkness at 28 °C. After 2 days of preconditioning, the rac-GR24 (1 × 10^−7^ mol/L) was then transferred into 24-well plate to induce seed germination for 3 days. After seed germination, rac-GR24 was removed and auxin IAA (1 × 10^−4^ mol/L), which can induce haustorium formation, was transferred to the 24-well plate and induced in darkness for 10 days [14]. Record radicle morphological changes, such as length and width. Radicle morphology hardly changes after haustorium formation, so the best time selection period is when radicle morphology stops changing.

### 2.3. Protoplasts’ Isolation and Purification

The protoplasts’ isolation and purification were modified based on Poddar and Yang’s protocols [15,16]. Approximately 0.1 ± 0.05 g of *P. aegyptiaca* seedlings with haustorium were placed in a 10 mL sterile tube and 5 mL of enzymatic solution was added to it. The sterile tube was then placed in a shaker at 23–27 °C and shaken at 80 rpm for 2–6 h away from light to remove the cell walls and release protoplasts. The enzymatic solution consists of 0.1%BSA, 10 mmol/L CaCl_2_, 0.1%MES and specific concentrations of mannitol, cellulase R-10 (YAKULT, Tokyo, Japan) and macerozyme R-10 (YAKULT, Japan); the concentrations of the latter three are shown in Table 1. In order to optimize the conditions of the enzymatic solution and improve the quality of protoplasts, the time and temperature of the enzymatic solution and the centrifugation of protoplasts are shown in Table 1. Therefore, the six factors and five levels orthogonal experiment (L25(5^6^)) was used, which contains 25 treatments. 

For purification, the enzyme lysate was filtered twice through a 70 μm cell filter to remove larger tissues. Subsequently, the filtrate was centrifuged for 5 min by the centrifugal force in Table 1 to remove the supernatant. The protoplast pellets at the bottom were then washed 3 times with 5 mL W5 solution (154 mmol/L NaCl, 125 mmol/L CaCl_2_, 5 mmol/L KCl, 5 mmol/L glucose, 2 mmol/L MES). Finally, the washed protoplast pellets were gently resuspended with 1 mL MMG solution (15 mmol/L MgCl_2_, 4 mmol/L MES, 0.5 mol/L mannitol) and placed on ice for 30 min for subsequent protoplast yield and viability evaluation and transient transformation execution. The protoplasts’ isolation and purification process of *P. aegyptiaca* is described in Figure 1.

### 2.4. Protoplasts’ Yield and Viability Assessment

The protoplasts’ yield was determined using a light microscope (Olympus, Tokyo, Japan, BX61). The protoplasts’ yield was calculated as follows: Protoplasts’ yield (protoplasts/g FW) = number of the protoplasts yielded in enzymolysis/fresh weight of the materials.

The protoplasts’ viability was determined by double staining with 2 μM Calcein-AM (Beyotime, Haimen, China) and 10 μM PI (Biorigin, Beijing, China) for 10 min at 37 °C in the dark. After staining, these were centrifuged for 5 min, washed with PBS once, and 100 μL PBS suspended protoplasts was added after washing. Then, the protoplasts were observed under a fluorescence microscope (Zeiss, Oberkochen, Germany, Axio Imager M2), and Calcein-AM reacted with living cells to fluoresce green under excitation light of 488 nm, while dead cells fluoresce red under excitation light of 535 nm under the action of PI. Protoplasts’ viability was calculated as follows: protoplasts’ viability (%) = (number of protoplasts with green fluorescence/total number of observed protoplasts) × 100%.

All the experiments were repeated three times. The process of determining the yield and viability of *P. aegyptiaca* protoplasts is described in Figure 1.

### 2.5. Protoplasts’ Transformation

PEG-mediated transient transformation in *P. aegyptiaca* haustorium protoplasts was performed based on the previously reported PEG-mediated transient transformation of kiwifruit and eggplant protoplasts [17,18] with slight modifications. The transformation process of *P. aegyptiaca* is described in Figure 1.

The fresh protoplasts were added to 2.5 × 10^5^ protoplasts/mL with MMG solution. Then, 200 μL protoplasts was taken and placed in a 2 mL tube. The pCAMBIA3301-35S-eGFP plasmid DNA and PEG solution (PEG4000, 0.5 mol/L mannitol, 0.1 mol/L CaCl_2_) were immediately added to it and the solution was gently mixed. To optimize plasmid content and PEG concentration, different amounts of plasmid (5, 10, 20, 30, 40 μg) and multiple concentrations of PEG4000 (20%, 30%, 40%, 50%, 60% *w*/*v*) were assayed (The PEG solution was filter sterilized using a 0.22 μm membrane filter). The mixture was then incubated at a specific temperature for a few minutes. To optimize incubation temperature and time, different temperatures (18, 21, 24, 27, 30 °C) and times (10, 20, 30, 40 min) were tested. After incubation, 1 mL W5 solution was slowly added to the mixture to stop the transformation. Then, the mixture was centrifuged at 100× *g* for 5 min to remove the liquid supernatant, and the remaining protoplast pellets were washed 2 times with W5 solution. The washed protoplasts were gently resuspended with 200 μL W5 solution. The transformed protoplasts were cultured in the dark at 28 °C for 12 h and then observed under fluorescence microscope. The GFP can be observed in the transfected protoplasts under a ZEISS Axio Imager M2 fluorescence microscope (excitation 488 nm). Transformation efficiency was calculated as follows:The transfection efficiency (%) = (the number of fluorescent protoplasts/the total number of protoplasts) × 100%.

All the experiments were repeated three times and five fields were observed for each replicate.

### 2.6. Detection of pCAMBIA3301, NPTII and eGFP in the Transfected Protoplasts by PCR

The transfected protoplasts’ and nontransfected protoplasts’ (control) total genomic DNA of *P. aegyptiaca* haustorium was extracted by using the Super Plant Genomic DNA Kit (TIANGEN). The presence of pCAMBIA3301 (pCAMBIA3301-*NPTII*-*eGFP*) in transfected protoplasts and its absence in nontransfected protoplasts was confirmed using pCAMBIA3301 primer (forward: 5′-CAGGAAACAGCTATGAC-3′; reverse: 5′-GTAAAACGACGGCCAGT -3′); *NPTII* primer (forward: 5′-AACTCACGTTAAGGGATTTTGGTCAT-3′; reverse: 5′-TCTTGGGGTATCTTTAAATACTGTAGAAAAGAGGA-3′) and *eGFP* primer (forward: 5′-GGTACCCGGGGATCCTCT-3′; reverse: 5′-GAAAGCTCTGCAGGAATTCGATT-3′).

PCR condition: [94 °C (3 min) [94 °C (30 s) 58 °C (30 s) 72 °C (1 min)] × 35 cycles 72 °C (5 min)]. PCR products were analyzed by 1% (*w*/*v*) agarose gel electrophoresis and ethidium bromide stain.

### 2.7. Statistical Analysis

The significant differences between different treatments were analyzed using the Tukey’s HSD test method at a significance level of *p* ≤ 0.05 by SPSS Version 22.0 (SPSS, Inc., Chicago, IL, USA). Conducting range analysis on orthogonal test results, the range value (R value) represents the quality of the factor’s effect on the experimental results. LSD multiple comparison analysis is used to assess the effects of 6 factors with 5 levels on the isolation and viability of protoplasts.

## 3. Results

### 3.1. Selection of Optimal Materials for Protoplasts’ Isolation of P. aegyptiaca Haustorium

After preconditioning, *P. aegyptiaca* seeds germinate white radicles under the induction of rac-GR24, and the radicles differentiate into haustoria under the induction of auxin. The morphological changes in the radicle during auxin-induced haustorium formation in *P. aegyptiaca* include the termination of radicle elongation, radial expansion of the meristematic region, and finally the epidermal cells at the haustorium initiation site form protrusions (Figure 2A–D) [5]. Therefore, radicles induced by auxin were selected for observation, and the length and width of the radicle at different times were compared (Figure 2E,F). In general, radicle elongation almost stopped under the induction of auxin, which mainly showed expansion of the meristematic region. By the 7th day, many protrusions formed on the radicle surface, and the radicle width reached a peak, after which the expansion of meristematic region stopped (Figure 2G,H). Thus, we suggested that haustoria were induced by applying auxin for 7 days after removing the rac-GR24 and were the optimal material for protoplasts’ isolation from *P. aegyptiaca* haustoria.

### 3.2. Selection of Optimal Factors for Protoplasts’ Isolation of P. aegyptiaca Haustorium

The results of the orthogonal experiment showed significant differences in the yield and activity of protoplasts across 25 different treatments. Among these, treatment 20 had the lowest protoplasts’ yield and activity, with 3.58 × 10^6^ p/g FW and 66.48%, respectively, and treatment 18 had the highest yield and activity, with 7.41 × 10^6^ p/g FW and 82.12%, respectively (Figure 3 and Figure 4 and Appendix A). Therefore, we concluded that protoplasts’ enzyme conditions, including the concentration of cellulase and macerozyme, the concentration of mannitol, the time and temperature of enzymolysis, and the centrifugation conditions, influence protoplasts’ isolation from *P. aegyptiaca* haustorium.

The range analysis results of protoplasts’ yield and activity showed that factors such as cellulase and macerozyme concentration, mannitol concentration, enzymolysis time and temperature, and centrifugal force have varying influences on the yield and viability of protoplasts’ isolation from *P. aegyptiaca* haustorium (Table 2). Among the six factors influencing protoplasts’ yield, the most influential is enzymolysis time, followed by Cellulase R-10, temperature, and mannitol concentration (Table 2). However, for protoplasts’ viability, the influence sequence of the six factors was mannitol > Macerozyme R-10 > Cellulase R-10 > enzymolysis temperature > centrifugal gravity > enzymolysis time (Table 2). The influence of different factors on the protoplasts’ yield and viability is different. Cellulase R-10, Macerozyme R-10 and mannitol have obvious influence on the protoplasts’ viability, while enzymolysis time, enzymolysis temperature and centrifugal gravity have great influence on the protoplasts’ yield.

To further determine the optimal levels of each factor of protoplasts’ isolation, we performed LSD multiple comparison analysis of yield and activity at each level of the factors (Figure 5). We found that protoplasts’ yield increased with the concentration of cellulase R-10, while the protoplasts’ activity decreased as the concentration of Cellulase R-10 increased. Moreover, the results in Table 2 show that the effect of Cellulase R-10 on protoplasts’ activity is much greater than on yield. Therefore, the maximum concentration of cellulose R-10 should be 2.5% to ensure high protoplasts’ activity. The effect of Macerozyme R-10 on protoplasts’ activity is similar to that of Cellulase R-10, but its impact on protoplasts’ yield is not as significant. Therefore, the maximum concentration of Macerozyme R-10 before a rapid decline in protoplasts’ activity should be 0.8%. The optimal mannitol concentration and enzymolysis time for protoplasts’ isolation are 0.5 mol/L and 4 h, respectively. In addition, at an appropriate temperature, the temperature has little effect on the activity of protoplasts. However, when the enzymolysis temperature is below or above 24 °C, the protoplasts’ yield is lower. Therefore, 24 °C is the optimal enzymolysis temperature. In conclusion, we determined that the optimal protocol for isolating protoplasts in the haustorium of *P. aegyptiaca* is 2.5% cellulase R-10, 0.8% Macerozyme R-10, 0.5 mol/L mannitol, enzymolysis temperature at 24 °C, enzymolysis time for 4 h, and centrifugation at 100× *g*.

### 3.3. Assessment of the Optimal Protocol in Protoplasts’ Isolation of P. aegyptiaca Haustorium

The optimal enzymolytic combination obtained by orthogonal test was used to separate the *P. aegyptiaca* haustorium protoplasts, and the protoplasts’ yield and viability were measured. The protoplasts’ yield was 6.2 × 10^6^ p/g FW, and the viability was 87.85% (Figure 6). The protoplasts obtained from the haustorium are described in Figure 6C–E, where most of the haustorium tissue is enzymatically decomposed into round protoplasts. The viability is illustrated by staining the protoplasts with Calcein-AM and PI (Figure 6F–N). The yield and viability of protoplasts obtained from the optimal combination were compared with those obtained from treatment 18 and treatment 20, respectively. We found that the yield and viability of protoplasts under the optimal combination were significantly higher than those under treatment 20 (Figure 6B). Therefore, the optimal enzymolytic combination obtained by the orthogonal test can isolate the *P. aegyptiaca* haustorium protoplasts with high yield and high viability.

### 3.4. Optimization of the Transient Transformation System of P. aegyptiaca Haustorium Protoplasts

Genetic factors can be transferred into protoplasts mediated by PEG, and this method has been widely used for transient expression in many plants [19]. In order to find an efficient transformation method of haustorium protoplasts, four factors affecting PEG-mediated haustorium protoplasts’ transformation, including temperature, incubation time, PEG concentration and plasmid content, were studied in this experiment. 

As shown in Figure 7, temperature and incubation time are important factors affecting transformation. The optimal temperature and time for the transformation of the haustorium protoplasts are 24 °C and 20 min, with transformation efficiencies of 58.26% and 64.4%, respectively. The concentration of PEG4000 is also a key factor in PEG-mediated protoplasts’ transformation. In this study, when the concentration of PEG4000 is 40%, the transformation efficiency of haustorium protoplasts is highest at 63%. This concentration is consistent with the one used by Shao et al. for the transformation of *Uncaria rhynchophylla* protoplasts [19]. In addition, the ratio of plasmids to protoplasts also greatly affects the transformation efficiency of protoplasts. When adding 5 μg of plasmids to 200 μL of protoplasts’ suspension, the transformation rate is only 49.58%, while adding 30 μg of plasmids increases the transformation rate to 78.49%.

In summary, the optimal scheme for protoplasts’ transformation is to add 200 mL 40% PEG4000 and 30 μg plasmid into 200 μL protoplasts’ suspension at 24 °C and incubate for 20 min. The transformation effect is shown in Figure 7H–J, with a transformation rate of 78.49%.

### 3.5. Assessment of the Optimized Protocol in Terms of Transfection Effect in Protoplasts of P. aegyptiaca Haustorium

We detected pCAMBIA3301, NPTII and *eGFP* genes in protoplasts transfected with pCAMBIA3301 plasmid by PCR and 1% agarose gel electrophoresis, while its absence was observed in nontransfected protoplasts (Figure 8), indicating that under the action of PEG4000, pCAMBIA3301 plasmid successfully integrated into the protoplasts of the haustorium from *P. aegyptiaca*.

## 4. Discussion

Plant protoplasts are living cells without cell walls surrounded only by cell membranes, which can directly absorb foreign DNA and are ideal receptors for transient gene expression [20]. Compared to stable gene expression, transient gene expression does not involve a regeneration process. After gene transformation, fluorescence signals can be directly observed, with a short period and high transformation efficiency [21]. Therefore, transient gene expression in protoplasts is particularly suitable for studies on subcellular localization, cell fusion, protein–protein interaction, promoter activity, and gene function analysis [9,22,23,24,25,26]. In this paper, we proposed a highly efficient method of protoplasts’ isolation and transient transformation of the haustorium from *P. aegyptiaca*. The protoplasts’ yield of the haustorium from *P. aegyptiaca* (6.2 × 10^6^ P/g FW) is much higher than that of mangoes (6.5 × 10^5^ P/g FW) isolated by Adjei [27]. The transformation efficiency (78.5%) in this paper was significantly improved compared to the transfection efficiency of citrus’ (68.4%) protoplasts reported recently [28].

It is worth noting that high-quality protoplasts are a prerequisite for the transient expression of protoplasts. However, haustorium is the main parasitic organ in *P. aegyptiaca;* to date, there is still no method to isolate its protoplasts. Although the haustorium of *P. aegyptiaca* is derived from the embryonic root, its structure is very different from that of the embryonic root. The epidermis of the haustorium contains Papillae cells and intrusive cells, and the interior contains a unique xylem bridge structure. The quality of protoplasts of the haustorium obtained by enzyme desorption of plant root protoplasts is poor. Therefore, although many methods for isolating protoplasts from plant roots have been reported, there is no effective method for isolating haustorium protoplasts. Moreover, there are many factors that affect the isolation of protoplasts, such as the type and concentration of enzymes, osmotic pressure of the enzyme solution, enzyme digestion time, enzyme digestion temperature, and even the magnitude of centrifugal force, all of which can influence the yield and activity of protoplasts. Ester’s research shows that a combination of 0.5% Cellulase R-10 concentration and 0.1% Macerozyme R-10 is sufficient to isolate high-yield and high-activity cabbage protoplasts [29], while Du’s research showed that the optimal concentrations of Cellulase R-10 and Macerozyme R-10 for digesting *Apium graveolens* protoplasts are 2.0% and 0.1%, respectively. Furthermore, it was found that once the yield and activity of *Apium graveolens* reached their highest values, increasing the enzyme concentrations actually resulted in a decrease in both the yield and activity of protoplasts [30]. These results indicate that, apart from the varying enzyme concentrations required for the digestion of different plant cell walls, the production and activity of protoplasts are not directly proportional to enzyme concentration, instead increasing with an increase in enzyme concentration and then declining. Our study found that the isolation of the haustorium of *P. aegyptiaca* protoplasts follows the same pattern. The optimal concentrations for isolating haustorium protoplasts are 2.5% cellulase R-10 and 0.8% Macerozyme R-10. However, when the cellulase R-10 concentration exceeds 2.5% or the Macerozyme R-10 concentration exceeds 0.8%, the protoplasts’ yield and activity decrease with the increasing enzyme concentration. Furthermore, although mannitol is a commonly used osmotic stabilizer in the isolation of protoplasts, its concentration is not fixed. A 0.4 mol/L mannitol concentration is the optimal concentration for isolating protoplasts from various plant species, including conifer callus, poplar, oak, and tea plant [31,32]. However, this study shows that when isolating haustorium protoplasts, the protoplasts’ activity and yield are highest when the mannitol concentration is 5 mol/L, which is the same as the optimal mannitol concentration for isolating carnation protoplasts [33]. Simultaneously, temperature affects the enzyme activity during protoplasts’ enzymatic digestion, thus influencing the rate of enzymatic reactions. A digestion time that is too short may not fully generate protoplasts, while a digestion time that is too long can damage the protoplasts’ cell membrane. Excessive centrifugal force can also lead to protoplasts’ rupture. Therefore, this study also investigated the effects of temperature, digestion time, and centrifugal force on the yield and activity of haustorium protoplasts. It was found that 24 °C and 4 h was the optimal enzymatic digestion temperature and time for isolation of haustorium protoplasts, and the quality of the protoplasts was not affected when the centrifugal force was lower than 100× *g*.

The currently common transient expression methods include particle biolistic bombardment, Agrobacterium-mediated transformation, and polyethylene glycol (PEG)-mediated protoplasts’ transformation [34,35,36]. Among them, the PEG-mediated protoplast transient gene transformation has become a rapid and effective tool for transient expression due to its ease of operation, high transformation efficiency, low cost, and no requirement for special equipment [12]. However, the difficulty of PEG-mediated protoplasts’ transient transformation in the haustorium of *P. aegyptiaca* lies in the lack of reference, and the parameters for PEG-mediated protoplasts’ transformation vary among different plants, and even among different organs of the same plant. Existing studies suggest that the main factors affecting PEG-mediated protoplasts’ transformation efficiency are incubation temperature, incubation time, PEG concentration, and plasmid content. Research by Zhang et al. demonstrated that the highest transfection efficiency (80%) for cotton root protoplasts was achieved when using 20 μg of plasmid in a 40% PEG4000 solution for 20 min [13]. Shao’s study suggests that for the *Uncaria rhynchophylla* protoplasts under conditions of 40% PEG4000 concentration, a plasmid concentration of 40 μg, and an incubation time of 40 min are required to achieve the maximum transformation efficiency [19]. Hyunhee’s study suggests that the optimal PEG4000 concentration for the transfection of *Camellia oleifera* petal protoplasts is 20%. Meanwhile, excessively high concentrations of PEG4000 can lead to protoplasts’ rupture [20]. Therefore, we investigated four factors affecting PEG-mediated protoplasts’ transformation and revealed the optimal parameters for PEG-mediated protoplasts’ transformation in the haustorium of *P. aegyptiaca*. We found that under 24 °C conditions, by adding 200 mL of 40% PEG4000 and 30 μg of plasmid to 200 μL of protoplasts’ suspension, and incubating for 20 min, the transformation efficiency of protoplasts in the haustorium from *P. aegyptiaca* could reach 78.49%.

Recently, some studies have proposed that protoplasts’ fusion can be used to prevent and control *Orobanche*. For example, Afshin enhances the ability of *Fusarium oxysporum* to inhibit *Orobanche* growth by fusing the protoplasts of *Fusarium oxysporum* and *Fusarium equina* [37]. However, there have been no reports on the prevention and treatment of *Orobanche* by using *Orobanche*’s own protoplasts. The method for high-yield and high-activity *P. aegyptiaca* haustorium protoplasts’ isolation proposed in this study, along with an efficient protoplasts’ transformation method, makes transient expression in *P. aegyptiaca* feasible. Transient expression in *P. aegyptiaca* provides a pathway for the study of haustorium functional genes. Studying the functional genes of haustorium can provide new avenues for the control of *P. aegyptiaca*. Therefore, the high-yield and high-activity *P. aegyptiaca* haustorium protoplasts’ isolation method and efficient protoplasts’ transformation method proposed in this study will contribute to advances in *P. aegyptiaca* disease control. Although extensive research on protoplasts has led to reported root, stem, and leaf protoplasts’ isolation and transformation methods for various model plants and major staple crops such as Arabidopsis, rice, and maize, the isolation and transformation of *P. aegyptiaca* haustorium protoplasts is first reported in this study. Moreover, the isolation and transformation of protoplasts from haustoria, a specific organ in parasitic plants, is also reported for the first time in this study, which can provide a reference for protoplasts’ isolation and transformation of haustorium in other parasitic plants. In recent years, single-cell sequencing in plants has become popular. The basis of plant single-cell sequencing relies on a large quantity of highly active plant protoplasts. The yield and activity of the *P. aegyptiaca* haustorium protoplasts isolated in this study meet the requirements for single-cell sequencing. Therefore, the protoplasts’ isolation method provided by this study will, to some extent, contribute to the development of single-cell sequencing in parasitic plants.

## 5. Conclusions

In summary, a reliable and efficient system for the isolation and transient gene expression of *P. aegyptiaca* haustorium protoplasts was developed. To the best of our knowledge, this is the first report of protoplasts’ isolation and PEG-mediated protoplasts’ transfection in *P. aegyptiaca* haustorium. This protocol provides technical support for studying functional genes and specific signal regulatory mechanisms in the haustorium of *P. aegyptiaca*. Additionally, since the haustorium is the key organ for parasitism, this protocol not only provides a new material for the study of *P. aegyptiaca*, but also provides a reference for the isolation and transformation of haustorium protoplasts in other parasitic plants.

## Figures and Tables

**Figure 1 plants-13-02163-f001:**
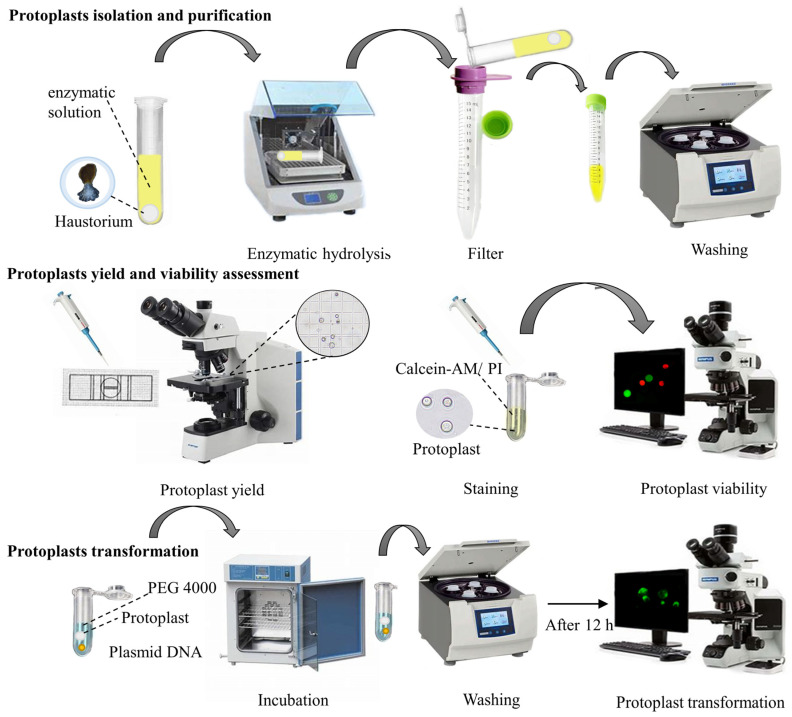
Protoplasts’ isolation and transformation procedures of the *P. aegyptiaca* haustorium.

**Figure 2 plants-13-02163-f002:**
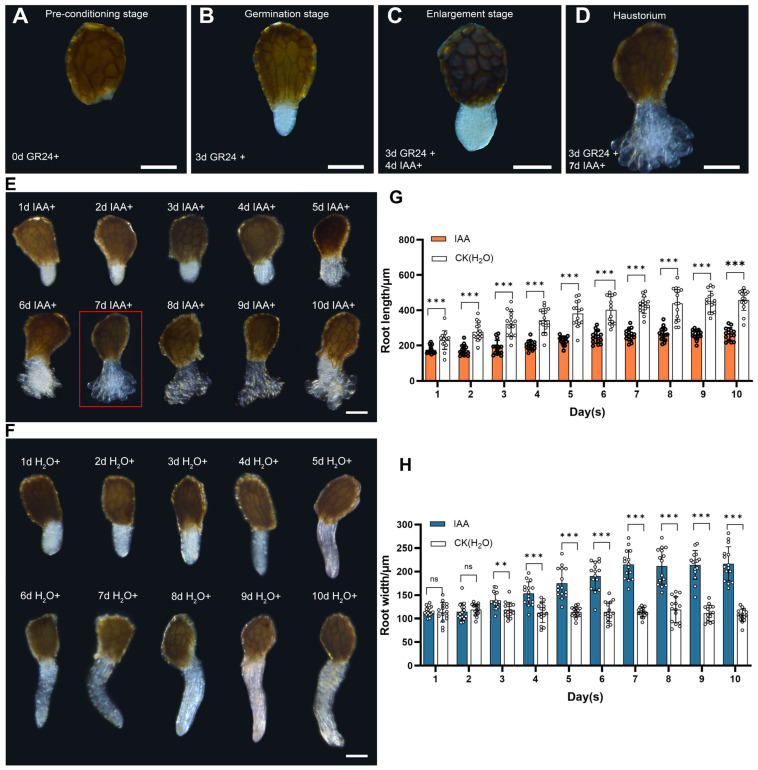
The process through which *P. aegyptiaca* seeds germinate (**A**,**B**) and form haustorium (**C**,**D**) morphological changes in radicle during auxin induction *P. aegyptiaca* haustorium formation. Scale bars = 200 μm. Morphological changes in seed radicle (**E**) at 10 days after auxin treatment. Morphological changes in seed radicle (**F**) at 10 days after H_2_O treatment (CK). Scale bars = 200 μm. The seed radicle was 10-day length (**G**) and width (**H**) under IAA and H_2_O treatment. Data presented are means ± SE. ns, **, *** indicate significant differences at *p* > 0.05, *p* < 0.01, *p* < 0.001, respectively, by Least Significance Difference Tests (LSDTs).

**Figure 3 plants-13-02163-f003:**
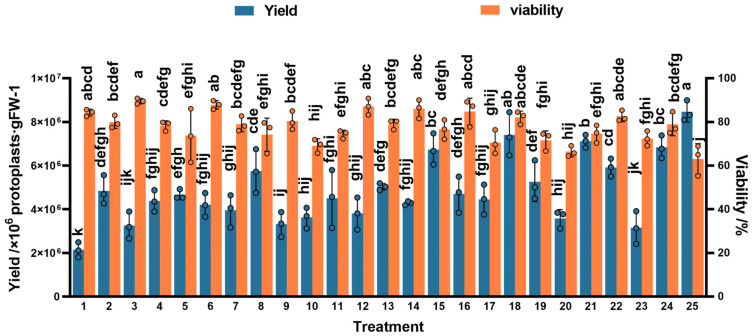
The results of protoplasts’ yield and protoplasts’ viability in the orthogonal experiment. The height of the rectangle indicates mean ± SE, and different letters indicate significant differences, *p* < 0.05.

**Figure 4 plants-13-02163-f004:**
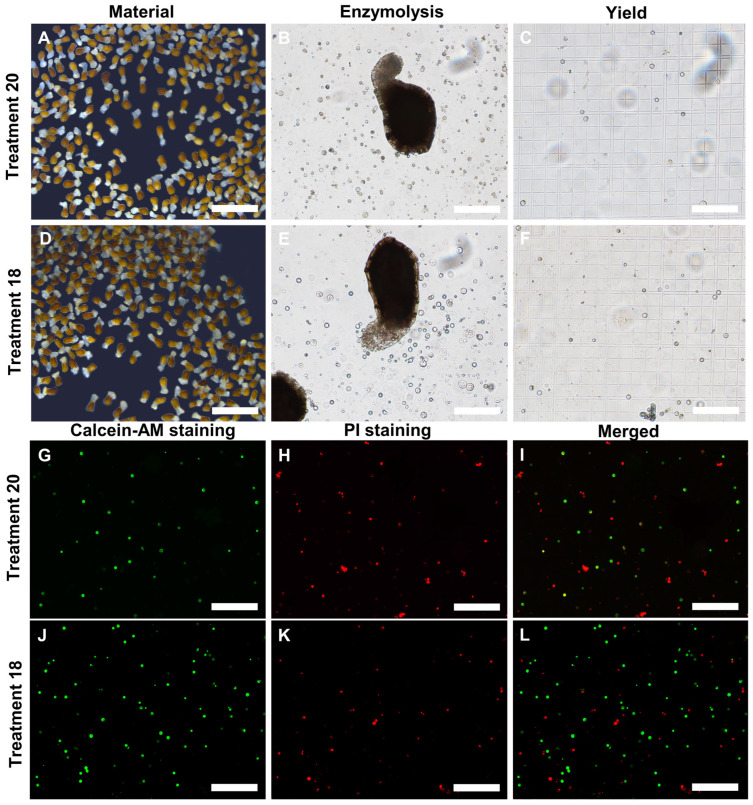
The yield of protoplasts obtained by *P.aegyptiaca* haustorium under treatment 20 (**A**–**C**) and treatment 18 (**D**–**F**). The protoplasts’ viability under treatment 20 (**G**–**I**) and treatment 18 (**J**–**L**) was examined by Calcein-AM (**G**,**J**) and PI (**H**,**K**) double staining. Green denotes living cells. Red denotes dead cells. All the bars are 200 μm except (**A**,**D**); (**A**,**D**) bars = 1000 μm.

**Figure 5 plants-13-02163-f005:**
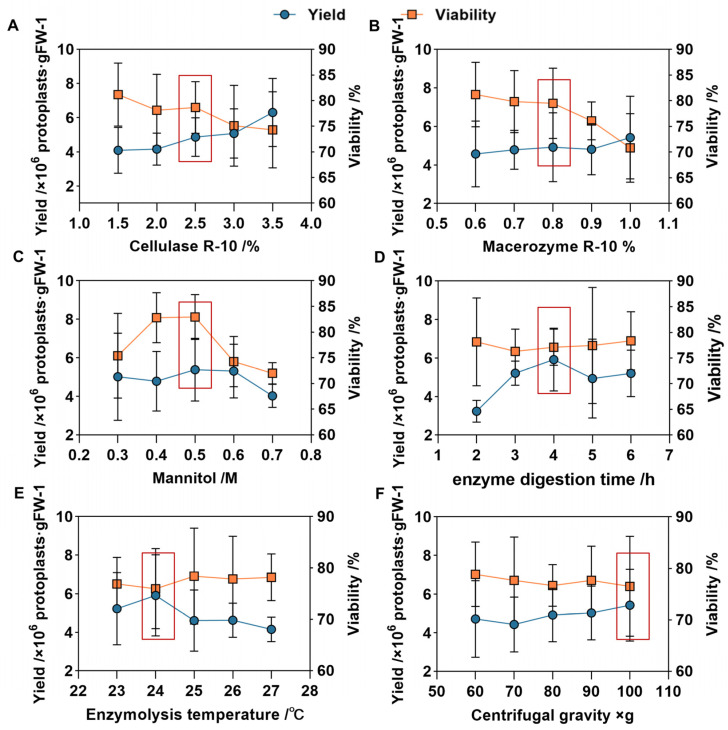
The effect of cellulase R-10 concentration (**A**); macerozyme R-10 concentration (**B**); mannitol concentration (**C**); enzymolysis time (**D**); enzymolysis temperature (**E**) and centrifugation (**F**) on protoplasts’ isolation from *P. aegyptiaca* haustorium.

**Figure 6 plants-13-02163-f006:**
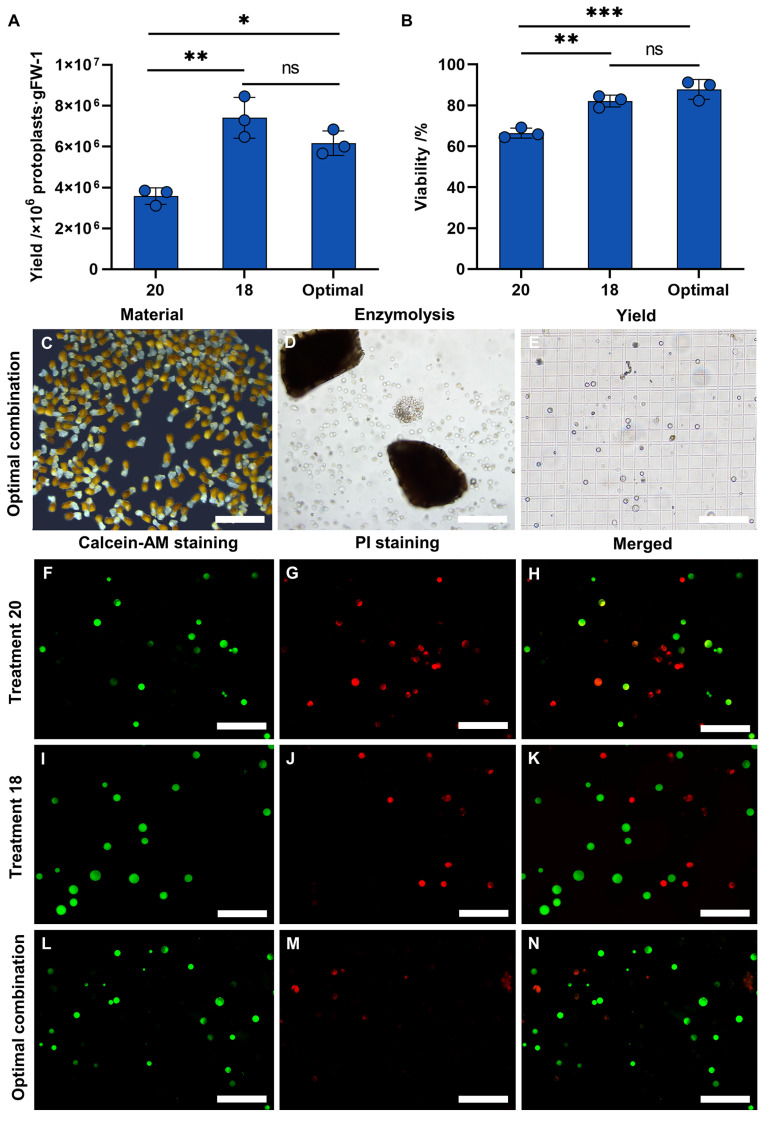
Protoplasts’ isolation in *P. aegyptiaca*; yield (**A**) and viability (**B**) of protoplasts’ isolation from the *P. aegyptiaca*. Data presented are means ± SE. ns, *, **, *** indicate significant differences at *p* > 0.05, *p* ≤ 0.05, *p* < 0.01, *p* < 0.001, respectively, by Least Significance Difference Tests (LSDTs). The protoplasts’ yield (**E**) of *P.aegyptiaca* (**C**) after enzymatic hydrolysis (**D**) under optimal combinations. The protoplasts’ viability under treatment 20 (**F**–**H**), treatment 18 (**I**–**K**) and optimal combinations (**L**–**N**) was examined by Calcein-AM (**G**,**J**) and PI (**H**,**K**) double staining. Green denotes living cells. Red denotes dead cells. (**C**) bar = 1000 μm; (**D**,**E**) bars = 200 μm; (**F**–**N**) bars = 100 μm.

**Figure 7 plants-13-02163-f007:**
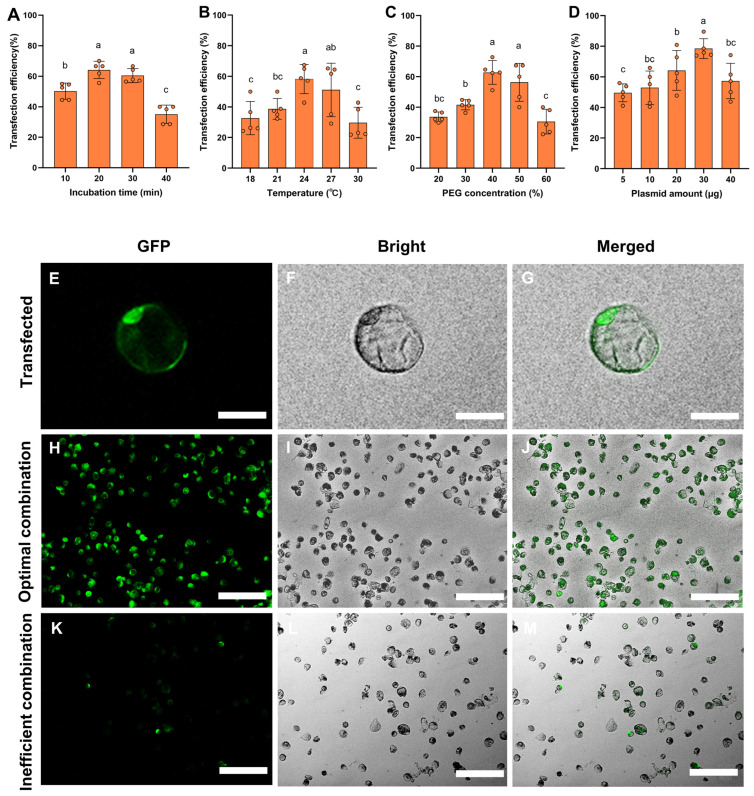
The effect of incubation time (**A**), temperature (**B**), PEG4000 concentration (**C**) and plasmid amount (**D**) on protoplasts’ transformation from *P. aegyptiaca* haustorium. The height of the rectangle indicates mean ± SE, and different letters indicate significant differences, *p* < 0.05. PEG-mediated transient gene expression in *P. aegyptiaca* haustorium protoplasts (**E**–**M**). The green fluorescent protein (GFP) gene expressed by 35S-eGFP vector can be observed in the transformed protoplasts under fluorescence microscope. (**E**–**G**) bars = 25 μm; (**H**–**M**) bars = 100 μm.

**Figure 8 plants-13-02163-f008:**
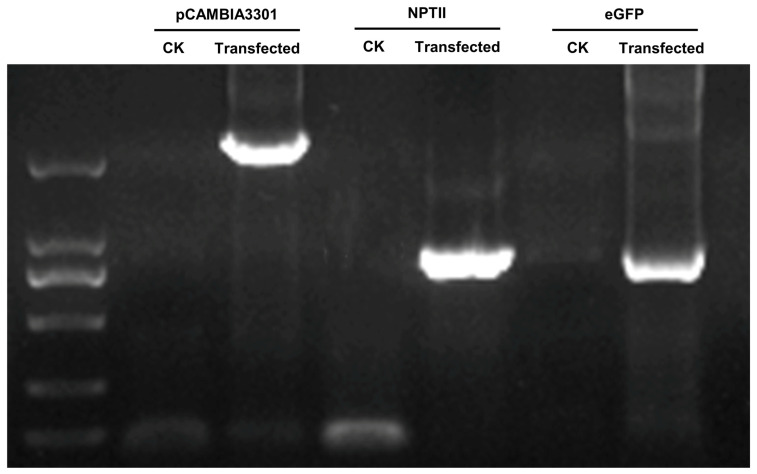
Detecting the presence of pCAMBIA3301, NPTII and *eGFP* in the protoplasts of *P. aegyptiaca* transfected with pCAMBIA3301—NPTII-eGFP. CK, nontransfected protoplasts.

**Table 1 plants-13-02163-t001:** Factor and level table.

Factor	Cellulase R-10 (%)	Macerozyme R-10 (%)	Mannitol (mol/L)	Enzymolysis Time (h)	Enzymolysis Temperature (°C)	Centrifugal Gravity (× *g*)
Level 1	1.5	0.6	0.3	2	23	60
Level 2	2	0.7	0.4	3	24	70
Level 3	2.5	0.8	0.5	4	25	80
Level 4	3	0.9	0.6	5	26	90
Level 5	3.5	1	0.7	6	27	100

**Table 2 plants-13-02163-t002:** The range analysis results of orthogonal experiments.

yield	K1	4.09	4.57	5.01	3.24	5.22	4.71
	K2	4.17	4.79	5.69	5.21	5.91	4.42
	K3	4.87	4.92	5.38	5.92	4.61	4.92
	K4	5.08	4.81	5.31	4.94	4.62	5.03
	K5	6.31	5.42	4.02	5.20	4.15	5.43
	Range	2.22	0.85	1.67	2.68	1.76	1.00
	Rank	enzymolysis time > Cellulase R-10 > enzymolysis temperature > mannitol > centrifugal gravity > Macerozyme R-10					
viability	K1	81.18	81.20	75.40	78.15	76.92	78.86
	K2	78.13	79.83	82.80	76.30	75.98	77.67
	K3	78.65	79.48	82.95	77.12	78.42	76.68
	K4	75.12	76.07	74.26	77.46	77.87	77.67
	K5	74.31	70.81	71.97	78.35	78.20	76.50
	Range	6.87	10.39	10.98	2.05	2.44	2.36
	Rank	mannitol > Macerozyme R-10 > Cellulase R-10 > enzymolysis temperature > centrifugal gravity > enzymolysis time					

## Data Availability

All data are included in the present study.

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
