# Peer review of "Isolation of Haustorium Protoplasts Optimized by Orthogonal Design for Transient Gene Expression in Phelipanche aegyptiaca"

_plants, 2024, doi:10.3390/plants13152163_

Round 1

Reviewer 1 Report

Comments and Suggestions for Authors

Dear authors, I have review your manuscript entitled:

Isolation of haustorium protoplasts optimized by orthogonal design for transient gene expression in Phelipanche aegyptiaca

Authored by

Xiao jian Zeng , Xiao lei Cao , Qiu yue Zhao , Si yuan Hou , Xin Hu , Zhe yu Yang , Ting li Hao , Si feng Zhao *, Zhao qun Yao *

Your manuscript is well written, but requires edition.  In the introduction you focused in protopast isolation and transient gene expression in this parasitic plant as an excelent approach to manipulate it. There is information about stable  genetic transformation and plant regeneration in this plant with no need of protoplasts. Also extensive information about defense against this parasitic plants mediated by genome editing.

You proposed in the introduction: We propose an advanced
and efficient transformation protocol for haustorium protoplasts from P. aegyptiaca.   You have to explain what  advanced knowledge is described your protocol.

Author Response

Comments 1: In the introduction you focused in protopast isolation and transient gene expression in this parasitic plant as an excelent approach to manipulate it. There is information about stable genetic transformation and plant regeneration in this plant with no need of protoplasts. Also extensive information about defense against this parasitic plants mediated by genome editing.

Response 1: Thank you for your valuable comments, we regret that our expression is not clear enough, affecting the reading. We chose to study the protoplasts of this parasitic plant for two main reasons: first, protoplasts are used to investigate gene expression at the single-cell level in this parasitic plant, providing a crucial platform for understanding its gene regulatory mechanisms in depth; second, we aim to establish a transformation and regeneration system under host-free conditions through protoplast research to enhance the flexibility of our studies and applications. Although we did not elaborate on other methods in the introduction, we believe that the use of protoplasts is the most suitable choice for our research objectives. We have added reasons to the introduction of the manuscript (line 49) to ensure that the reader has a more complete understanding of the field.

Comments 2: You proposed in the introduction: We propose an advanced and efficient transformation protocol for haustorium protoplasts from P. aegyptiaca.  You have to explain what advanced knowledge is described your protocol.

Response 2: We are very grateful to you for your professional comments on our article. According to your suggestions, we summarized the following three advanced knowledge in our article. Advanced Knowledge 1: Protoplast separation of Gualedan haustorium was first proposed in this paper, but has not been reported before. Advanced Knowledge 2: The transient protoplast transformation system of Phelipanche aegyptiaca was first proposed in this paper, and has not been reported before. Advanced Knowledge 3: The transient expression system of Phelipanche aegyptiaca haustorium protoplast established in this paper has a high transformation efficiency, which provides material for the study of gene function of Phelipanche aegyptiaca.

Comments 3: Line 29 genus.

Response3: Thank the reviewers for their valuable suggestions. We have added this part according to your suggestions.

Comments 4: Line 56 scientific name in italic.

Response4: Thank you very much for your suggestion. According to your suggestion, we have changed the scientific name of line 56 to italics.

Comments 5: Lines 84-87 Which auxin and how much mg/L

Response 5: Thank you for your careful examination. We are sorry for our carelessness. According to your suggestion, we have indicated the type and concentration of auxin in the paper. The type of auxin used is IAA, and the concentration used is 1×10-4 mol/L.

Comments 6: Line 142 Do the PEGs solution were filter sterilized? Otherwise, if you autoclave it a different final concentration occurred in your experiments.

Response 6: We appreciate your professional comments on our article. The PEG solution we used was sterilized by filtration. Thanks for your suggestion, I marked the article with filter sterilization.

Comments 7: Kanamycin is proper name for the antibiotic, but you used the neomycin phosphotransferase type 2 gene, NPTII. Please used it instead of Kana.

Response 7: Thank you for your professional advice. According to your suggestion, we have checked the full text and replaced the kana in the full text with NPTII.

Comments 8: References are not written in the style of mdpi.

Response 8: We sincerely thank the reviewers for their careful reading. We are very sorry for the errors in the format of the references and have corrected them.

Reviewer 2 Report

Comments and Suggestions for Authors

Zeng and co-workers presented a study about the optimized Isolation and transient gene expression in protoplasts of Phelipanche aegyptiaca.

It is a linear work. Not very innovative but with some interest and well done. Personally I’d be interested to see it published, despite it may be considered a limited technical advancement.

The main concern is about figures.

No one single protoplast is shown at high magnification. To see the shape of protoplasts before and after transient expression is important to understand their quality. The low magnification images show stressed cells. Demonstrating they are in good shape/health is essential. High magnification images are necessary and I consider it a major revision. 

In addition there are minor points regarding editing. For example:

Line 56 scientific name in italic.

Lines 84-87 check soundness

Line 189. I understand that authors have good reasons to select that material but this is not necessarily optimal. Authors decided it is optimal. Please rephrase.

Lines 213-214. Legend for A-F is not clear.

Line 217. Correct “Bare”

Figure 7: It would be better to show a picture with few transformants but not zero.

Lines 393-3: Please explain with some details the cited reference.

Comments on the Quality of English Language

some editing is required, see general comments

Author Response

Comments 1: No one single protoplast is shown at high magnification. To see the shape of protoplasts before and after transient expression is important to understand their quality. The low magnification images show stressed cells. Demonstrating they are in good shape/health is essential. High magnification images are necessary and I consider it a major revision. 

Response 2: Thank you very much for the professional advice of the reviewers. We have thought carefully about your suggestion and have added a high magnification protoplast image to the Figure 7 section of the article.

Comments 2: Line 56 scientific name in italic.

Response 2: Thank you very much for your suggestion, According to your suggestion, we have changed the scientific name of line56 to italics.

Comments 3: Lines 84-87 check soundness

Response 3: Thank you for your careful examination. We are sorry for our carelessness. According to your suggestion, we have indicated the type and concentration of auxin in the paper. The type of auxin used is IAA, and the concentration used is 0.175mg/L.

Comments 4: Line 189. I understand that authors have good reasons to select that material but this is not necessarily optimal. Authors decided it is optimal. Please rephrase.

Response 4: I really appreciate your comments. We have carefully considered your question and added changes in the article.

Comments 5: Lines 213-214. Legend for A-F is not clear.

Response 5: Thanks for Reviewer’s suggestion, we have modified and marked in the text.

Comments 6: Line 217. Correct “Bare”

Response 6: Thank the reviewers for their careful examination. Sorry for our careless, we have changed “bars” into “bare”.

Comments 7: Figure 7: It would be better to show a picture with few transformants but not zero.

Response 7: Thank you for your professional advice. According to your advice, we have modified picture 7 and replaced it with a more suitable picture.

Comments 8: Lines 393-3: Please explain with some details the cited reference.

Response 8: Thank you for your valuable comments. According to your advice, we have explained the cited reference in detail and marked them in the article.

Round 2

Reviewer 1 Report

Comments and Suggestions for Authors

Dear authors I have read the recommendations I have asked you. The manuscript is ready.

Author Response

Thanks to the reviewers for their valuable comments, which made this article improved

Reviewer 2 Report

Comments and Suggestions for Authors

I appreciate the revision. 

Comments on the Quality of English Language

Please reveise "Bare" into "Bars"

Author Response

Comments 1: Please reveise "Bare" into "Bars"

Response 1: Thank you very much for your suggestion, we have made changes in the article.